# GABA_B_R Modulation of Electrical Synapses and Plasticity in the Thalamic Reticular Nucleus

**DOI:** 10.3390/ijms222212138

**Published:** 2021-11-09

**Authors:** Huaixing Wang, Julie S. Haas

**Affiliations:** Department of Biological Sciences, Lehigh University, 111 Research Drive, Bethlehem, PA 18015, USA; wanghuaixing2000@yahoo.com

**Keywords:** gap junction connexin36, LTD, GABA_B_ receptor

## Abstract

Two distinct types of neuronal activity result in long-term depression (LTD) of electrical synapses, with overlapping biochemical intracellular signaling pathways that link activity to synaptic strength, in electrically coupled neurons of the thalamic reticular nucleus (TRN). Because components of both signaling pathways can also be modulated by GABA_B_ receptor activity, here we examined the impact of GABA_B_ receptor activation on the two established inductors of LTD in electrical synapses. Recording from patched pairs of coupled rat neurons in vitro, we show that GABA_B_ receptor inactivation itself induces a modest depression of electrical synapses and occludes LTD induction by either paired bursting or metabotropic glutamate receptor (mGluR) activation. GABA_B_ activation also occludes LTD from either paired bursting or mGluR activation. Together, these results indicate that afferent sources of GABA, such as those from the forebrain or substantia nigra to the reticular nucleus, gate the induction of LTD from either neuronal activity or afferent glutamatergic receptor activation. These results add to a growing body of evidence that the regulation of thalamocortical transmission and sensory attention by TRN is modulated and controlled by other brain regions. Significance: We show that electrical synapse plasticity is gated by GABA_B_ receptors in the thalamic reticular nucleus. This effect is a novel way for afferent GABAergic input from the basal ganglia to modulate thalamocortical relay and is a possible mediator of intra-TRN inhibitory effects.

## 1. Introduction

The thalamic reticular nucleus (TRN) is a generator of sleep spindle rhythms [1] and is thought to act as a possible focusing mechanism for a cortical ‘searchlight’ of attention [2,3]. The TRN functions as a gate for the bidirectional flow of activity and information between the thalamus and the cortex. The TRN receives its major source of excitatory input from topographically organized as well as diffuse cross-modal collaterals of both thalamocortical and corticothalamic axons. The synaptic targets of inhibition from the TRN are the thalamic relay neurons [4,5]. Within the TRN, connexin36-based electrical synapses are the main substrate for neuronal communication [6]; TRN neurons communicate amongst themselves via ionotropic GABAergic synapses mainly up to early adolescence in rats [7]. Thus, the electrical synapses in the TRN, and their strength, are likely key regulators of brain rhythms and attention.

Two types of activity are known to induce depression at electrical synapses of the TRN. Burst firing in TRN, a pattern of regular sodium spikes crowning longer T-channel calcium spikes, is a prominent component of both sleep spindle rhythms and the sharp wave discharges that characterize *absence* seizures [8,9,10,11,12]. When two electrically coupled TRN neurons are made to fire synchronized bursts, the electrical synapse connecting them undergoes LTD [13]. Tetanization of afferent cortical glutamatergic input to coupled TRN neurons also drives LTD that is mediated by the recipient mGluRs [14]. In mGluR-dependent LTD, induced by bath application of the agonist ACPD, group I and group II mGluRs modulate coupling through opposing effects: activation of group I mGluRs works through a Gs signaling pathway, stimulating adenylyl cyclase, activating PKA and resulting in depression, while activation of the group II receptor mGluR3 yields potentiation through activation of Gi/o, adenylyl cyclase and inhibition of PKA [15]. The burst-induced form of plasticity depends on calcium influx through voltage-gated channels that leads to intracellular calcium release and phosphatase activation, while the ACPD-induced plasticity is independent of calcium influx. Together, these results demonstrate that electrical synapse strength is modulated quite specifically by incoming and intrinsic forms of neuronal activity.

Up to 30% of synapses onto TRN neurons are GABAergic [16,17]. There are several known sources of GABAergic inputs to the TRN, including projections from the substantia nigra reticulata [18] and the basal forebrain [19,20,21]. TRN also receives GABAergic projections from the globus pallidus [22,23,24] and hypothalamus [25]. While ionotropic intra-TRN GABAergic connectivity is known to sharply diminish after early adolescence [Hou et al. 2016], metabotropic intra-TRN connectivity remains uncharacterized.

GABA_B_ receptors are strongly expressed in TRN [26,27]. GABA_B_ receptors mediate diverse pre- and postsynaptic effects. Activation of GABA_B_ receptors inhibits adenylyl cyclase via the Gi/o subunits of G proteins, resulting in upregulation of the inward-rectifier potassium (GIRK) current, modulation of T current activation, and reduction of burst firing in TRN neurons [28]. GABA_B_ further modulates calcium entry by high-voltage activated channels, thereby regulating the presynaptic release of neurotransmitters. Activation of GABA_B_ receptors is also known to inhibit the calcium-activated potassium current [29], which is prominent in TRN neurons. Likely through their effect on PKA, GABA_B_ receptors have been shown to both modulate and be modulated by the processes underlying chemical synaptic plasticity [30]. Because mGluR-mediated LTD of electrical synapses is initiated by G proteins, adenylyl cyclase and PKA activation [15], and because mGluR-mediated and activity-dependent forms of electrical synapse LTD occlude each other [31], we sought to understand whether GABA_B_ receptor activation would modulate the two known initiators of LTD.

## 2. Results

We first tested the effects of GABA_B_ a receptor activity on electrical synapse strength alone. Inactivation of GABA_B_ receptors by the antagonist CGP 55845 (10 µM) resulted in a modest but significant depression of electrical synapse strength (Figure 1A–D; ΔG_C_: −6.1 ± 0.5%, *p_w_* = 0.005; Δcc = −4.6 ± 0.6%, *p_w_* = 0.03, *n* = 18 pairs) with no change in input resistance (Figure 1E). Resting membrane potential, −68.9 mV, was unchanged (*p_w_* = 0.6, *n* = 36 neurons) by CGP application.

Activation of GABA_B_ receptors by the agonist R-baclofen (Figure 2; 10 µM) lowered input resistance by from 323.1 ± 27.8 MΩ to 303.3 ± 27.7 MΩ % (*p_w_* = 0.001, *n* = 14 neurons) in our data, as expected from activating GIRK channels [28,32], together with a 1.8 mV decrease in resting membrane potential (*p* = 0.01). Coupling was also modulated by baclofen: G_C_ decreased by 7.6 ± 1.1% (*p_w_* = 0.03) and cc decreased by 12.4 ± 0.9% relative to the control (*p_w_* = 0.001, *n* = 15 pairs). Together, these two results indicate that electrical synapse strength is regulated at baseline by GABAergic inputs to TRN.

Electrical synapses can be depressed by paired bursting activity, as we have shown previously [13]. We repeated that paradigm here (Figure 3A–D), which resulted in depression in conductance by 16.9 ± 0.8% (*p_w_* = 0.004) and depression in coupling by 10.8 ± 1.4% (*p_w_* = 0.007, *n* = 8 pairs). As previously, input resistance was not changed by induced activity.

We examined the effects of LTD-inducing paradigms in the presence of GABA_B_ receptor modulation. Inactivation of GABA_B_ receptors by exposure to CGP (20 min prior to bursting) prevented LTD induction by paired bursting activity (Figure 4A–D; ΔG_C_ = −2.2 ± 1.0%, *p_w_* = 0.57; Δcc = −1.3 ± 0.8%, *p_w_* = 0.30 (*n* = 12 pairs), with no effect on input resistance. The application of ACPD (50 µM) following the application of CGP also failed to induce LTD (Figure 5A–D; ΔG_C_ = −1.5 ± 1.0%, *p_w_* = 0.5, and Δcc = 4.2 ± 1.1%, *p_w_* = 0.2, relative to values after CGP wash-in), also with no effect on input resistance.

As baclofen induced strong depression of electrical synapse strength (Figure 2), we expected that it would also occlude further depression induced by paired bursting or by ACPD. Our results showed that was the case; performing paired bursting or the application of ACPD after 20 min of pre-stimulus exposure of the slice to baclofen failed to induce further changes in synapse strength (Figure 6; ΔG_C_ = −0.4 ± 1.8%, *p_w_* = 0.4; Δcc = −1.2 ± 1.1%, *p_w_* = 0.5, *n* = 8 pairs). ACPD application also failed to induce LTD (Figure 7; ΔG_C_ = −1.2 ± 1.1%, *p_w_* = 0.38; Δcc = −0.58 ± 0.97%, *p_w_* = 0.65, *n* = 19 pairs). Together, these results also indicate that some baseline activation of GABA_B_ receptor activity, but no more, is required in order for the successful induction of LTD, in addition to setting its baseline strength.

We directly compared LTD induced by paired bursting in the control ACSF to the effects of paired bursting in agonist or antagonist by unpaired t-tests of normalized coupling values after activity. These confirmed that activity in ACPD failed to induce the same effect as bursting in ACSF (*p* = 0.038 for cc, *p* = 0.002 for G_C_), and similar for CGP (*p* = 0.043 for cc, *p* = 0.017 for G_C_).

Our findings are summarized by the pathway proposed in Figure 8. Briefly, we observed that GABA_B_ receptor activity acts both as a requirement and a gate for LTD induction of electrical synapses. We also propose that the antagonization of GABA_B_ receptors by CGP inhibits GABAergic regulation of calcium influx through the T channel pathway [28,33] and modulation of bursting dynamics by the calcium-activated potassium current; together these effects prevent the amount of calcium influx required for burst-induced LTD. We suggest that ACPD-induced plasticity shifts towards its underlying LTP mechanisms in the absence of GABA_B_ activation. On the other hand, activation of GABA_B_ receptors by baclofen precludes depression induced by mGluR tetanization or by induced bursts, possibly by saturating PKA [15]. Further characterization of these specific interactions remains to be investigated.

## 3. Discussion

Together, our results support the role of GABA_B_ receptors in both establishing the baseline strength and gating the plasticity of electrical synapses. These results add to an accumulation of evidence that electrical synapses and their plasticity mechanisms interact with other neurotransmitter systems [15,34,35,36,37]. In principle, many similar interactions between synaptic input, intracellular signaling pathways, intrinsic properties, and electrical synapse strength are possible across the brain, and remain to be elucidated. While our results indicate that the postsynaptic metabotropic pathways activated by glutamatergic and GABAergic receptors may overlap, we note that the nuclei and transmitters that activate them in the TRN and elsewhere are distinct and can thus exert distinct effects. Regulation by metabotropic activation of GABA receptors, as demonstrated here, could be one route for intra-TRN GABA to operate, and to regulate intra-TRN connectivity, activity and synchrony.

We suggest that our results reveal a complex interaction between GABA signaling in its many forms and electrical synapse strength. The burst-induced form of LTD depends on calcium influx through T- and high-voltage-activated channels [31], both of which are modulated by GABA_B_ receptors. Decreased activation of calcium-activated potassium channels may also result in decreased activation of T channels and resulting calcium influx below the amount necessary to induce LTD. Smaller amounts of calcium, on the other hand, produce LTP at these synapses [38], indicating a precise interaction between calcium influx and activity-dependent plasticity outcome. Our results also suggest that some baseline or tonic activation of GABA_B_ receptors is required for the correct calcium influx necessary to drive LTD and imply that LTD may be enabled only in that case. In addition to its various afferent sources, ambient GABA is an essential controller of TRN neurons [39], and GABA_B_ receptors are strongly expressed in TRN [26,27]. The necessary activation of GABA_B_ receptors for LTD could, in principle, occur tonically {Hung, 2020 #722} or by co-activation of GABAergic inputs to TRN during activity that results in LTD. Further experiments will be required to distinguish these possibilities, specifically identifying tonic activity of GABA_B_Rs in TRN, and to fully flesh out the calcium-based interactions between GABA_B_ and the plasticity that results from activity within TRN neurons.

ACPD-induced LTD is also a balanced outcome; in that case between two competing plasticity processes, activation of group I mGluRs leads to increased PKA and results in depression, while activation of the group II receptor mGluR3 yields potentiation through inhibition of PKA [15]. Our results here also indicate a balancing regulatory role for GABA_B_ receptor activation, where removing it entirely seems to shift the balance of ACPD induction towards its LTP mechanisms (Figure 5), possibly through G protein interactions, although the LTP was not significant in this case. Altogether, these results indicate that levels of GABA_B_ receptor activation distinctly regulate both the outcome to afferent activity (glutamate)-induced plasticity and plasticity that results from activity within TRN neurons. In both cases, LTD can be prevented by the baclofen saturation of PKA activation. We note that because we did not perform experiments in which plasticity induction was followed by GABA_B_ receptor modulators, our results do not fully address occlusion.

Some limitations to these findings should be noted. First, like all extant direct, paired studies of electrical synapses in the TRN to date [6,13,14,15,31,38,40,41,42,43,44,45,46], the brain slices used in our experiments were from juvenile, P11-P15 rats. After eye opening, myelinated fibers from relay thalamus occlude optics in the TRN and make dual recordings from nearby pairs of likely-coupled neurons much less feasible. However, the expression profiles of connexin36 indicate support that TRN electrical synapses at this age are likely representative of adult synapses [47,48,49], and TRN neurons have matured by this stage [50]. Extracellular evidence in the TRN shows that the hallmarks of strong electrical synapses persist into adulthood [51]. It remains possible that some aspects of plasticity are different at the synapses in fully developed animals. Advances in optics and in slice-preserving solutions may enable our future work to confirm this correspondence in synapses and plasticity across age groups and allow for an increased understanding of plasticity in adults and across development. Optogenetic experiments indicate that ionotropic GABA transmission is mostly lacking within adult TRN [7], but these results do not address metabotropic activation of TRN neurons by within-TRN sources, which remains a possibility; effects of baclofen similar to those shown here have been demonstrated in older TRN tissue [28,52]. However, GABAergic synapses in TRN appear to connect more distant neurons [53] but are absent between coupled nearby pairs [6].

Variability within our results could arise from variability in underlying mechanisms. The TRN is organized into sectors corresponding to its afferents [54,55,56]. Within TRN, there are distinct subtypes of TRN neurons [57,58,59], and varying spatial patterning of gap junction-coupled networks [42]. This specificity, and the variability within our data, raises the possibility of the varying effects of GABA_B_ mechanisms within distinct subtypes of TRN neurons. In particular, we saw a slight, though not significant, increase in coupling for ACPD applied in the presence of CGP. Because ACPD itself activates competing plasticity processes [15], these data might indicate that GABA could shift the balance between them, possibly in a cell-specific manner.

The magnitude of changes in electrical synapse strength we demonstrate is similar to that in our previous demonstration of LTD [13,31], well less than 20% (GC or cc). These magnitudes of plasticity are comparable to those previously shown for TRN electrical synapses via mGluR-induced LTD [14] and of the same order for modulation in the inferior olive [60,61,62]. The numerically modest changes in synaptic strength shown here yield 5–10 ms changes in spike times in coupled neighbors [63], and computational models have reinforced the functional diversity and effectiveness of changes in coupling of this magnitude [64,65].

Within TRN, electrical synapses are the dominant source of interneuronal communication. For that reason, modulation of their strength is of special interest to thalamocortical communication, where the TRN is thought to focus one possible instantiation for a searchlight of attention, out of many possible sources. Afferent input to TRN by external areas, such as the substantia nigra or the forebrain, represent diverse ways in which emotional or sensory salience could regulate cortical sensory attention. Modulation of synaptic connectivity adds to the repertoire of how input from other brain areas could modify thalamocortical communication. TRN is also a crucial nexus for the rhythms that underlie stage two of sleep and absence epilepsy; responsiveness of those conditions to treatment by GABA analogues may rely, in part, on modulation of electrical synapse and electrical synapse LTD, as shown by these experiments.

## 4. Methods

**Electrophysiology** Horizontal slices 3–400 μm thick were obtained from Sprague-Dawley rats aged P11–P14 of both sexes, in accordance with previous studies in TRN [13]. Rats were anesthetized by inhaled isofluorane (5 mL of isoflurane applied to fabric, within a 1 L chamber) and euthanized by decapitation, in accordance with federal and institutional university IACUC animal welfare guidelines. Slices were cut and incubated in sucrose solution (in mM): 72 sucrose, 83 NaCl, 2.5 KCl, 1 NaPO_4_, 3.3 MgSO_4_, 26.2 NaHCO_3_, 22 dextrose, 0.5 CaCl_2_. Slices were incubated at 37 °C for 20 min and returned to room temperature until recording. The bath for solution during recording contained (in mM): 126 NaCl, 3 KCl, 1.25 NaH_2_PO_4_, 2 MgSO_4_, 26 NaHCO_3_, 10 dextrose and 2 CaCl_2_, 300–305 mOsm, saturated with 95% O_2_/ 5% CO_2_. The submersion recording chamber was held at 34 °C (TC-324B, Warner Instruments, Hamden, CT, USA). Micropipettes were filled with (in mM): 135 K-gluconate, 2 KCl, 4 NaCl, 10 HEPES, 0.2 EGTA, 4 ATP-Mg, 0.3 GTP-Tris, and 10 phosphocreatine-Tris (pH 7.25, 295 mOsm). An amount of 1M KOH was used to adjust pH of the internal solution. The approximate bath flow rate was 2 mL/min, and the recording chamber held approximately 5 mL solution. Drugs applied were R-baclofen (10 µM), CGP 55845 (10 µM) and ACPD (50 µM) and were acquired from Tocris (Minneapolis, MN, USA) or Sigma (St. Louis, MO, USA) and diluted into high-concentration stock solutions in DMSO or water before final dilution. Final DMSO concentration was always <0.2%. Drugs were bath-applied continuously unless noted otherwise.

The TRN was visualized under 5× magnification, and pairs of TRN cells were identified with 40x IR-DIC optics (SliceScope, Scientifica, Uckfield, UK). Signals were amplified and low-pass filtered at 8 kHz (MultiClamp, Axon Instruments, Novato, CA, USA), digitized at 20 kHz (lab-written Matlab routines controlling a National Instruments USB6221 DAQ board), and stored for offline analysis in Matlab (Mathworks, R2012a, Natick, MA, USA). All recordings were made in whole-cell current-clamp mode. Values of V_rest_ ranged from −50 to −70 mV, and negative current was used to maintain all cells at −70 mV during measurement of coupling in order to accurately measure it [40,66,67]. Measurements of R_in_ and coupling were made by 100–300 pA of negative current injection. Pipette resistances were 4–8 MΩ before bridge balance, which was discarded if it exceeded 25 MΩ. Voltages are reported uncorrected for the liquid junction potential. During ACPD application, further negative current was added to prevent spiking that could become bursting and confound the induction stimuli we sought to separate, but neurons were allowed to depolarize to just below threshold. Paired bursting was induced by current injection (200–400 pA) through the recording electrodes for 50 ms at 2 Hz, as described in Haas et al. 2011 [13].

**Numerical analysis** Input resistances for each cell and coupling between cells were quantified by injecting 100 pA of hyperpolarizing current into one cell of a coupled pair, and measuring voltage deflections in that cell (ΔV) and in the coupled neighbor (δV). The coupling coefficient cc is computed as δV/ΔV. Coupling coefficients reported here are averaged over 10 measurements per timepoint in both directions, averaged together. Coupling conductance G_c_ was estimated separately for each direction from the same measurements [45,68], and also averaged over both directions. For plasticity experiments, experiments were discarded when R_in_ of either cell in a pair deviated by more than 20% from initial values, or when the coefficient of variation for coupling measurements exceeded 0.1. Slices were not randomly assigned to treatments, nor were investigators blinded. Changes in coupling were evaluated as the average over the 20 min following activity or drug application after 2 minutes of wash-in for drugs, compared to normalized baseline values, and are reported as mean ± SEM. We report two-sided Wilcoxon signed-rank tests on the sets of change in coupling for each condition, and outcomes are reported as *p_w_*. Power exceeded 0.8 for all comparisons. No multiple comparisons were performed.

## Figures and Tables

**Figure 1 ijms-22-12138-f001:**
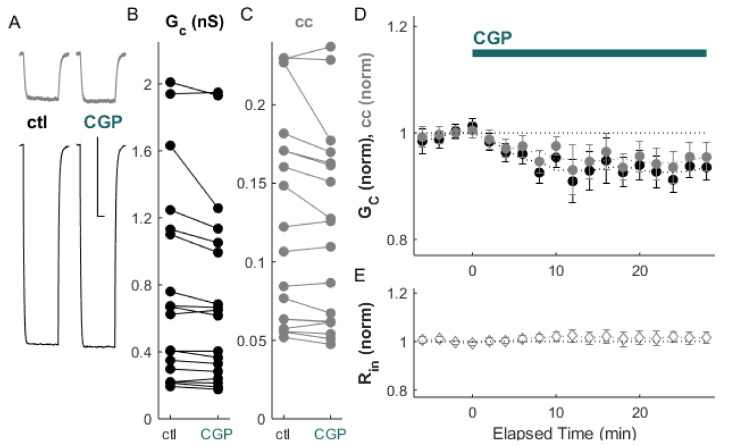
Coupling is depressed by the GABA_B_ antagonist CGP. (**A**) Example of paired recordings of electrically coupled neurons, in response to a −100 pA step to one neuron (bottom trace), before (ctl) and after bath CGP application. Scale bar 100 ms, 5 mV for both neurons. (**B**) Conductance G_C_ and (**C**) coupling coefficients cc for the cohort of pairs used in this experiment. (**D**) After CGP application, ΔG_C_ (black) was −6.1 ± 0.5%, *p_w_* = 0.005, and Δcc (grey) was −4.6 ± 0.6% relative to control, *p_w_* = 0.03 (*n* = 18 pairs). (**E**) Input resistance for all neurons over the experiment.

**Figure 2 ijms-22-12138-f002:**
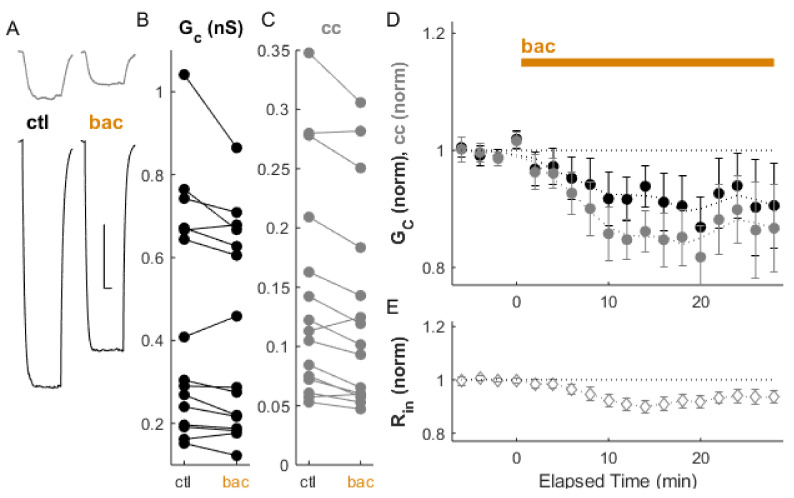
Coupling is depressed by the GABA_B_ receptor agonist baclofen. (**A**) Example of paired recordings of electrically coupled neurons, in response to a −100 pA step to one neuron (bottom trace), before (ctl) and after bath baclofen (bac) application. Scale bar 100 ms, 5 mV for both neurons. (**B**) Conductance G_C_ and (**C**) coupling coefficients cc for the cohort of pairs used in this experiment. (**D**) After baclofen application, ΔG_C_ (black) was −7.6 ± 1.1%, *p_w_* = 0.03, and Δcc (grey) was −12.4 ± 0.9% relative to control, *p_w_* = 0.001 (*n* = 15 pairs). (**E**) Input resistance for all neurons over the experiment.

**Figure 3 ijms-22-12138-f003:**
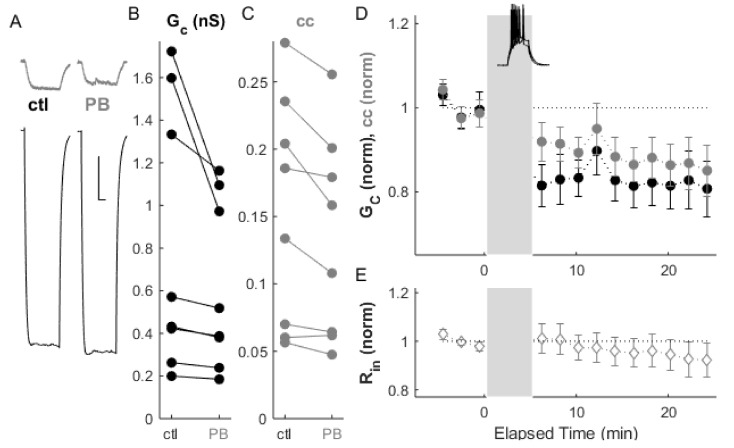
Bursts induce LTD of electrical synapses. (**A**) Example of paired recordings of electrically coupled neurons, in response to a −100 pA step in one neuron (bottom), before and after 5 min of paired bursting (PB, inset in **D**) activity in both neurons. Scale bar 100 ms, 5 mV (injected neuron), 10 mV (passive neuron). (**B**) Conductance G_C_ and (**C**) and coupling coefficients cc for the cohort of pairs used in this experiment. (**D**) After paired bursting, ΔG_C_ (black) was depressed by 16.9 ± 0.8%, *p_w_* = 0.004, and Δcc (grey) decreased by 10.8 ± 1.4%, *p_w_* = 0.007 (*n* = 8 pairs). (**E**) Input resistance for all neurons over the experiment.

**Figure 4 ijms-22-12138-f004:**
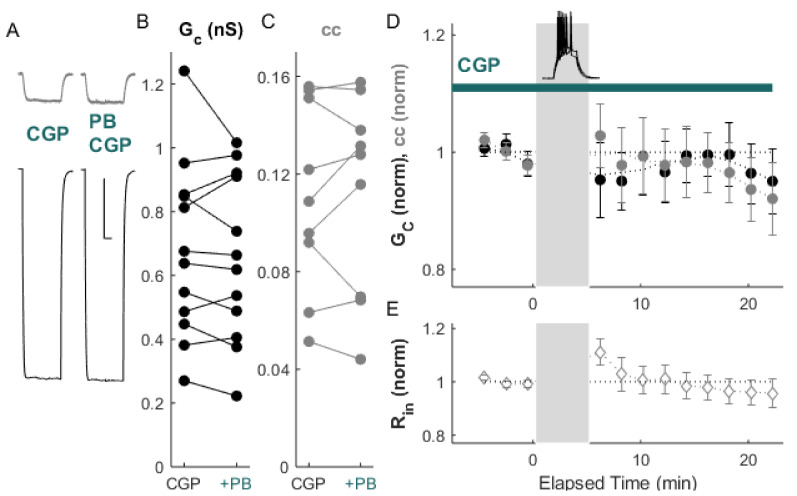
Bursts fail to induce LTD in the presence of the GABA_B_ receptor antagonist CGP. (**A**) Example of paired recordings of electrically coupled neurons, in response to a −100 pA step in one neuron (bottom), before and after 5 min of paired bursting (PB) with CGP in the bath. Scale bar 100 ms, 5 mV for both neurons. (**B**) Conductance G_C_ and (**C**) coupling coefficients cc for the cohort of pairs used in this experiment. (**D**) After paired bursting, ΔG_C_ (black) was −2.2 ± 1.0%, *p_w_* = 0.57, and Δcc (grey) was −1.3 ± 0.8% relative to CGP controls, *p_w_* = 0.30 (*n* = 12 pairs). (**E**) Input resistance for all neurons over the experiment.

**Figure 5 ijms-22-12138-f005:**
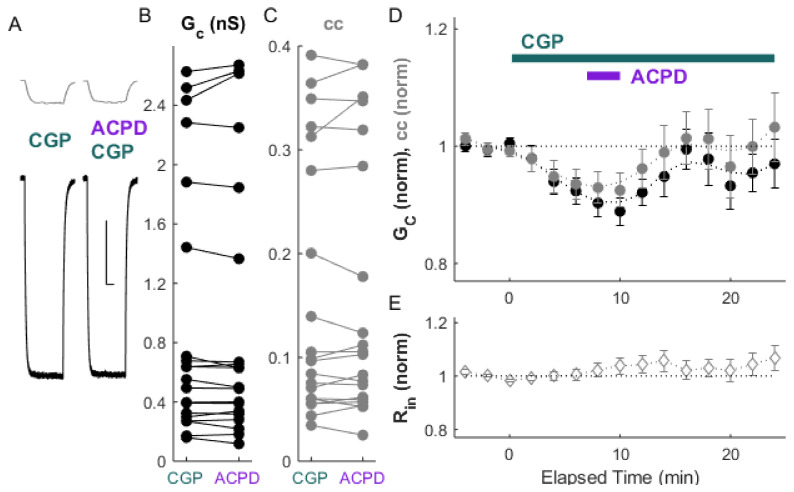
ACPD fails to induce LTD in the presence of the GABA_B_ receptor antagonist CGP. (**A**) Example of paired recordings of electrically coupled neurons, in response to a −100 pA step to one neuron (bottom trace), before and after bath ACPD application, with CGP in the bath. Scale bar 100 ms, 5 mV for both neurons. (**B**) Conductance G_C_ and (**C**) and coupling coefficients cc for the cohort of pairs used in this experiment. (**D**) After ACPD application (50 µM), ΔG_C_ (black) was −1.5 ± 1.0%, *p_w_* = 0.5, and Δcc (grey) was 4.2 ± 1.1%, *p_w_* = 0.2, relative to values in CGP (between 2 and 7 min) (*n* = 20 pairs). (**E**) Input resistance for all neurons over the experiment.

**Figure 6 ijms-22-12138-f006:**
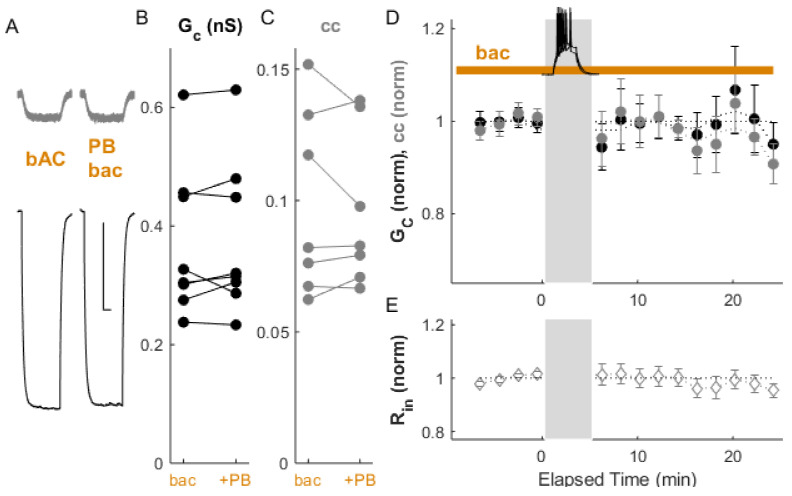
Paired bursting fails to induce LTD in the presence of the GABA_B_ receptor agonist baclofen. (**A**) Example of paired recordings of electrically coupled neurons, in response to a −100 pA step to one neuron (bottom trace), before and after paired bursting, with baclofen in the bath. Scale bar 100 ms, 5 mV for both responses. (**B**) Conductance G_C_ and (**C**) coupling coefficients cc for the cohort of pairs used in this experiment. (**D**) After paired bursting, ΔG_C_ (black) was −0.4 ± 1.8%, *p_w_* = 0.4, and Δcc (grey) was −1.2 ± 1.1%, *p_w_* = 0.5 from pre-stimulus control (*n* = 8 pairs). (**E**) Input resistance for all neurons over the experiment.

**Figure 7 ijms-22-12138-f007:**
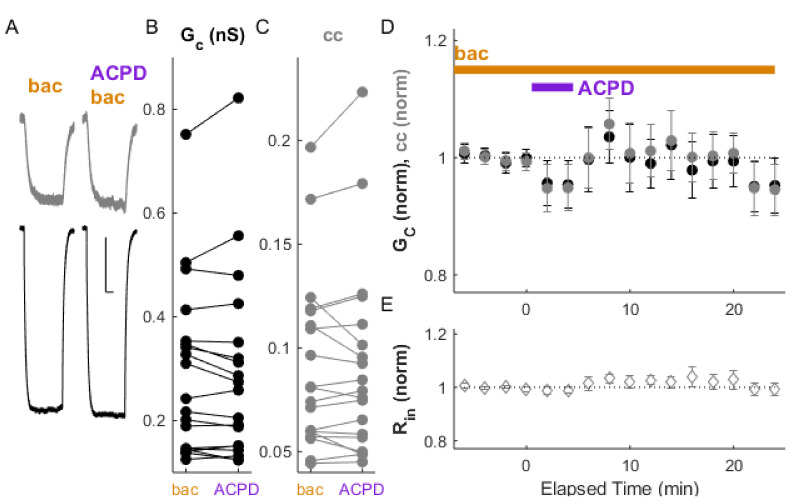
ACPD-induced LTD is blocked by the GABA_B_ receptor agonist baclofen. (**A**) Example of paired recordings of electrically coupled neurons, in response to a −100 pA step to the one neuron (to bottom neuron) before and after ACPD application, in the constant presence of baclofen. Scale bar 100 ms, 5 mV (injected neuron), 10 mV (passive neuron). (**B**) Conductance G_C_ and (**C**) coupling coefficients cc for the cohort of pairs used in this experiment. (**D**) After 4 min of ACPD exposure (50 µM), ΔG_C_ (black) was −1.2 ± 1.1%, *p_w_* = 0.38, and Δcc (grey) was −0.58 ± 0.97%, *p_w_* = 0.65 from pre-stimulus control (*n* = 19 pairs). (**E**) Input resistance for all neurons over the experiment.

**Figure 8 ijms-22-12138-f008:**
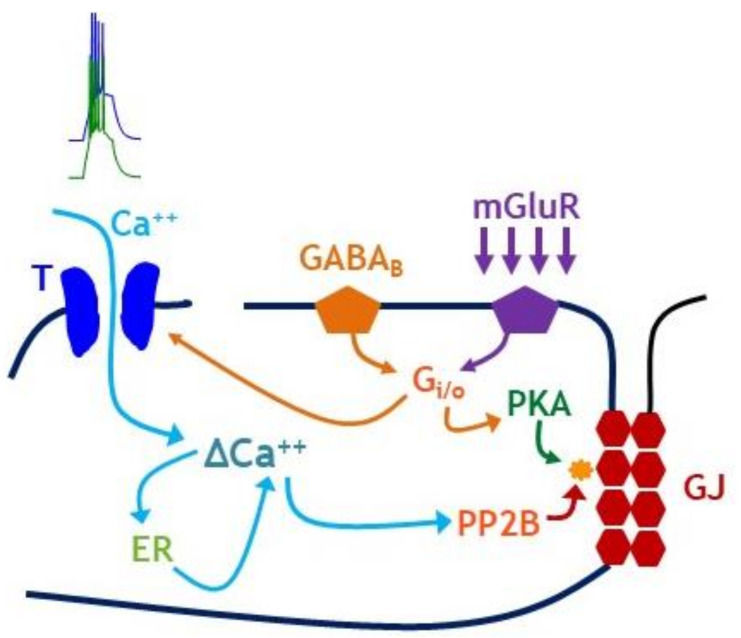
Proposed interactions between GABA_B_ receptors and established LTD pathways. LTD of electrical synapses can be induced by paired bursting (left; [31]) or by mGluR tetanization (right; [15]). Our results indicate that GABA_B_ receptors interacts with both burst-induced and mGluR-induced signaling pathways. We propose that burst-induced LTD requires GABA_B_ receptor activation to maintain its calcium dependence. ACPD-induced plasticity reverts to its underlying LTP mechanisms in the absence of GABA_B_ receptor activation. Burst-induced LTD and ACPD-induced LTD are both blocked by GABA_B_ receptor saturation of PKA at the endpoint of phosphorylation.

## Data Availability

Data are available from the corresponding author (J.S.H.) upon reasonable request.

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
