# Peer review of "GABABR Modulation of Electrical Synapses and Plasticity in the Thalamic Reticular Nucleus"

_ijms, 2021, doi:10.3390/ijms222212138_

Round 1
Reviewer 1 Report
This is a very tidy publication, and I support it's publication as is.
However, I do want to mention to the authors that they are committing a statistic sin. It is very common, (indeed there have been whole articles dedicated to this sin), but it is still naughty. In this case, I do not think it matters, but the authors might at least want to consider correcting it.
Specifically, if you have two interventions, and you are interested if they have a differential effect, one cannot perform a t-test on one intervention, and get a p<0.05 and then perform t-test on the other intervention and get p>0.05, and the conclude that these two interventions have different effects. You have literally never tested whether the magnitude of the effects of these interventions are different.
Concretely, around line 160 you show that in the presence of balofen, paired bursting had no detectable effect on CC. You then conclude that paired bursting has a different effect in the presence or absence of baclofen. Strictly speaking, this is not valid, as you never tested this.
There are several ways to point out the problem with this approach. What if one t test gave you p = 0.049 and one gave you p = 0.051. Would you say the effect was different then? What if you had huge power in one test, and no in the other?
There are several "correct" ways to approach this, and typically they involve something akin to a two way ANOVA, where you explicitly have an effect of bursting, and an effect of drug, and what you would expect to see is an interaction, i.e. the effect of bursting depends on the presence of drug. In this case, you would need to use a mixed effects model to deal with the paired nature of the data, but conceptually it is the same.
As I have said, I do not think this makes any difference to the conclusions drawn in the paper. But on the other hand, the authors still might want to consider updating their analysis.
Author Response
We thank the reviewers for their time and attention, and for their support of our work.
Changes to the manuscript are marked by blue text.
This is a very tidy publication, and I support it's publication as is.
However, I do want to mention to the authors that they are committing a statistic sin. It is very common, (indeed there have been whole articles dedicated to this sin), but it is still naughty. In this case, I do not think it matters, but the authors might at least want to consider correcting it.
Specifically, if you have two interventions, and you are interested if they have a differential effect, one cannot perform a t-test on one intervention, and get a p<0.05 and then perform t-test on the other intervention and get p>0.05, and the conclude that these two interventions have different effects. You have literally never tested whether the magnitude of the effects of these interventions are different.
Concretely, around line 160 you show that in the presence of balofen, paired bursting had no detectable effect on CC. You then conclude that paired bursting has a different effect in the presence or absence of baclofen. Strictly speaking, this is not valid, as you never tested this.
There are several ways to point out the problem with this approach. What if one t test gave you p = 0.049 and one gave you p = 0.051. Would you say the effect was different then? What if you had huge power in one test, and no in the other?
There are several "correct" ways to approach this, and typically they involve something akin to a two way ANOVA, where you explicitly have an effect of bursting, and an effect of drug, and what you would expect to see is an interaction, i.e. the effect of bursting depends on the presence of drug. In this case, you would need to use a mixed effects model to deal with the paired nature of the data, but conceptually it is the same.
As I have said, I do not think this makes any difference to the conclusions drawn in the paper. But on the other hand, the authors still might want to consider updating their analysis.
Thank you for your support, and for pointing out this omission. As you note, there are several ways to approach comparing results of bursting in control to those in baclofen and CGP. We have chosen to perform an unpaired t-test on the normalized average coupling values. These comparisons, which confirm that the outcome of bursting is different between control and drugs, are now reported in the Results.
Reviewer 2 Report
This manuscript provides the first evidence that the plasticity of electrical synapses in the neonatal/young thalamic reticular nucleus (in acute slices) can be modulated by the activity of GABAB receptors. Electrical synapses are perhaps the most important element of internal communication between reticular inhibitory neurons and are relatively understudied. These studies extend the body of knowledge about the mechanisms of plasticity of electrical synapses to the role of GABAB receptor. The presented results suggest that electrical synapses in reticular nucleus exhibit fine-tuned strength that is decreased by both pharmacological activation or inhibition of the GABAB receptor.
The manuscript by Wang and Haas describes the involvement of GABAB receptor in the plasticity of GABAergic synapses. Using electrophysiological approaches the Authors evaluated the effects of GABABR manipulation and found that activation or inhibition of this receptor contributes to the enduring plasticity of electrical synapses in a similar way. This work builds on prior evidence that electrical synapses of TRN undergo long-term depression upon the period of elevated and synchronized activity of excitatory synapses. In these studies, they evaluated effects of GABABR ligand or inhibitor to dissect the involvement of GABAB receptors in LTP of electrical synapses. All recordings were performed in acute brain slices prepared from neonatal/young (P11-14) rats. It is important to bear in mind that at this age GABAergic system can be still depolarizing, and reticular nucleus is further characterized by the significant presence of inhibitory chemical synapses, the number of which decreases considerably with the development.
Presented results show that interfering with GABAB receptor caused impairment in the LTD of electrical synapses. Application of GABABR agonist or antagonist, both leads to the induction of LTD in electrical synapses. Additionally, these forms of plasticity block further the induction of LTD by paired-bursting protocol or application of APCD, an mGluR agonist.
In all, this is an interesting paper, describing an intriguing data, which together makes a compelling case that inhibitory synaptic transmission (phasic or tonic) may modulate the strength of electric synapses. These are important studies describing entirely novel findings. The work should be of interest researchers working on mechanisms of synaptic plasticity and physiology of electrical synapses. Especially, knowing that some interneurons in the cortex or hippocampus are interconnected by gap-junction, the possibility that GABA can modulate the efficacy of their coupling is intriguing.
By in large the methods are well presented and the illustrations support the conclusions reached. The Authors do a good job considering the pertinent literature and discussing their results. I do have comments that call for some revisions to the manuscript, largely pertaining to the discussion, statistical analysis and interpretation of results. My specific concerns are listed below.
Major questions (may require additional experiments)
- To claim that the pharmacological interference in GABABR functioning occluded LTD induced by paired-bursting protocol or application of APCD, the manuscript needs the experiment in the opposite order: first induce LTD with paired-bursting protocol or application of APCD, and then test the effect of GABAB inhibitor or activator. Only when both stimuli prevent the induction by the other, one can interpret that they both OCCLUDED each other (because they share a common downstream signaling pathway for induction and expression). If not, the Authors should rewrite all statements about “occlusion”.
- The results presented in Figure 5 suggest that CGP pretreatment causes ACPD administration (which under control condition results in LTD induction) to potentiate the coupling between neurons, suggesting the induction of LTP of electric synapses. Authors should consider to perform additional experiments explaining this shift in plasticity direction under the blockade of GABAB Additionally, Authors should perform statistical analysis of data at time points right before and after ACPD infusion using paired statistical tests to check whether in the presence of CGP, the infusion of ACPD potentiates electrical synapses. This potentiating effect is already discussed in the manuscript , but it should primarily result from a detailed statistical analysis of the data.
- The Authors use the "tonic levels of GABA" in many places in the manuscript. As is known, the ambient concentration of GABA induces the tonic activity of the GABAA receptor. Do the Authors have this activity in mind when using the phrase "tonic levels of GABA"? If so, these statements should be reinforced by evidence that the tonic concentration of GABA can activate metabotropic GABAB receptors in TRN. If not, the terms “tonic GABA” and similar should be discontinued throughout the manuscript.
- The limitation of this work is the neonate/young age of the used rats. As the Authors themselves write, the number of inhibitory synapses in TNR decreases with age. The Authors should consider conducting at least one demonstration experiment showing that the observed effect of baclofen or CPG administration is also taking place in adult sections (at least P45). If not, I suggest strengthening the part of discussion about the age of isolated brain sections by pointing to papers showing strong GABAB transmission in the absence (or low level) of GABAA dependent transmission in other brain regions. Authors already mentioned this in discussion (line 235-237) without any citation confirming such possibility.
- The manuscript does not contain detailed information regarding the interval between the administration of baclofen and the application of the pairing protocol (Fig.6.) or ACPD (Fig. 7). It is not clear why in Figure 5D the start of CGP administration is shown, and on Figure 4D not. This figure will benefit if the depressing action of CGP will be visible before the pairing protocol.
- I am not sure if I understand correctly the Authors' intentions regarding statistical analysis. I found, that Authors use two statistical tests: the parametric and nonparametric one for the analysis of every dataset. It is quite confusing because the gold standard is to use a parametric test (e.g. t-test) for data with a normal distribution and a non-parametric test (e.g. Mann – Whitney U test) for data whose distribution is not normal. Therefore, I suggest that the Authors shall verify the distributions of the data (judging by the scatter charts, these distributions are usually not normal) and apply one appropriate test to assess the difference. Alternatively, Authors may use a nonparametric test for all data. I believe that reporting of the results of a parametric t-test for data whose distribution deviates from normal is a methodological error.
- The Authors should explicitly state if investigators used criteria for data inclusion and exclusion (e.g. outliers), if the slices were randomly assigned to treatments, if investigators were blinded to the treatments during data collection and/or analysis, how were sample numbers chosen, and the statistical Power yielded by the sample numbers used.
Author Response
To claim that the pharmacological interference in GABABR functioning occluded LTD induced by paired-bursting protocol or application of APCD, the manuscript needs the experiment in the opposite order: first induce LTD with paired-bursting protocol or application of APCD, and then test the effect of GABAB inhibitor or activator. Only when both stimuli prevent the induction by the other, one can interpret that they both OCCLUDED each other (because they share a common downstream signaling pathway for induction and expression). If not, the Authors should rewrite all statements about “occlusion”.
Thank you for pointing out this distinction. We have changed the word “occluded” in several instances to “prevented” or “blocked”. We have added text to the Discussion to clarify this distinction: We note that because we did not perform experiments in which plasticity induction was followed by GABAB modulators, our results do not fully address occlusion.
The results presented in Figure 5 suggest that CGP pretreatment causes ACPD administration (which under control condition results in LTD induction) to potentiate the coupling between neurons, suggesting the induction of LTP of electric synapses. Authors should consider to perform additional experiments explaining this shift in plasticity direction under the blockade of GABAB Additionally, Authors should perform statistical analysis of data at time points right before and after ACPD infusion using paired statistical tests to check whether in the presence of CGP, the infusion of ACPD potentiates electrical synapses. This potentiating effect is already discussed in the manuscript , but it should primarily result from a detailed statistical analysis of the data.
Thank you for noticing this interesting point. The statistics performed for Figure 5 do compare the timepoints in CGP, right before and after ACPD application; this is noted in both the Results and the Figure legend (“relative to values in CGP (between 2 and 7 min)”. The increase in coupling was not significant. Due to the lack of significant LTP, we limit our description of the Result to the lack of LTD induction for this section.
To provide more clarity, we altered out description in the Discussion to “Our results here also indicate a balancing regulatory role for GABAB activation, where removing it entirely seems to shifts the balance of ACPD induction towards its LTP mechanisms (Fig. 5), possibly through G protein interactions, although the LTP was not significant in this case.”
Due to the time limitations and extended number of additional experiments that would be necessary, we apologize that additional experiments are beyond the scope of the current manuscript, while they remain of interest to our group.
The Authors use the "tonic levels of GABA" in many places in the manuscript. As is known, the ambient concentration of GABA induces the tonic activity of the GABAA receptor. Do the Authors have this activity in mind when using the phrase "tonic levels of GABA"? If so, these statements should be reinforced by evidence that the tonic concentration of GABA can activate metabotropic GABAB receptors in TRN. If not, the terms “tonic GABA” and similar should be discontinued throughout the manuscript.
Thank you for pointing out the need for clarification here. We have reduced our use of the term ‘tonic’ to a single instance in the Discussion, which we hope communicates our ideas more specifically: Our results also suggest that some baseline or tonic activation of GABAB receptors is required for the correct calcium influx necessary to drive LTD, and imply that LTD may be enabled only in that case. In addition to its various afferent sources, ambient GABA is an essential controller of TRN neurons (Crabtree et al. 2013), and GABAB receptors are strongly expressed in TRN (Margeta-Mitrovic et al. 1999; Munoz et al. 1998). The necessary activation of GABAB receptors for LTD could in principle occur tonically {Hung, 2020 #722} or by co-activation of GABAergic inputs to TRN during activity that results in LTD. Further experiments will be required to distinguish these possibilities, specifically identify tonic activity of GABABRs in TRN, and to fully flesh out the calcium-based interactions between GABAB and plasticity that results from activity within TRN neurons.
The limitation of this work is the neonate/young age of the used rats. As the Authors themselves write, the number of inhibitory synapses in TNR decreases with age. The Authors should consider conducting at least one demonstration experiment showing that the observed effect of baclofen or CPG administration is also taking place in adult sections (at least P45). If not, I suggest strengthening the part of discussion about the age of isolated brain sections by pointing to papers showing strong GABAB transmission in the absence (or low level) of GABAA dependent transmission in other brain regions. Authors already mentioned this in discussion (line 235-237) without any citation confirming such possibility.
Thank you for this opportunity to clarify. As suggested, we have added a sentence to the Discussion: effects of baclofen similar to those shown here have been demonstrated in older TRN tissue (Cain et al. 2017; Zhang et al. 2020).
Due to the time limitations and the sheer difficulty of recording in adult TRN, we apologize that additional experiments are beyond the scope of the current manuscript, while they remain of interest to our group.
The manuscript does not contain detailed information regarding the interval between the administration of baclofen and the application of the pairing protocol (Fig.6.) or ACPD (Fig. 7). It is not clear why in Figure 5D the start of CGP administration is shown, and on Figure 4D not. This figure will benefit if the depressing action of CGP will be visible before the pairing protocol.
Thank you for this opportunity to clarify. We have added the following phrase to the Results in order to specify the intervals: … paired bursting performed or application of ACPD after 20 min of pre-stimulus exposure of the slice to baclofen. and a similar indication in the Results that describe pre-exposure to CGP in Figure 4. For GGP+ACPD (Fig. 5), there was no pre-exposure period. We have changed text to more clearly indicate these timings in the text.
I am not sure if I understand correctly the Authors' intentions regarding statistical analysis. I found, that Authors use two statistical tests: the parametric and nonparametric one for the analysis of every dataset. It is quite confusing because the gold standard is to use a parametric test (e.g. t-test) for data with a normal distribution and a non-parametric test (e.g. Mann – Whitney U test) for data whose distribution is not normal. Therefore, I suggest that the Authors shall verify the distributions of the data (judging by the scatter charts, these distributions are usually not normal) and apply one appropriate test to assess the difference. Alternatively, Authors may use a nonparametric test for all data. I believe that reporting of the results of a parametric t-test for data whose distribution deviates from normal is a methodological error.
Thank you for this opportunity to clarify. We have reverted to using the non-parametric two-sided Wilcoxen signed rank test, and report these values as pw.
The Authors should explicitly state if investigators used criteria for data inclusion and exclusion (e.g. outliers), if the slices were randomly assigned to treatments, if investigators were blinded to the treatments during data collection and/or analysis, how were sample numbers chosen, and the statistical Power yielded by the sample numbers used.
We have added these details to the Methods.
Round 2
Reviewer 2 Report
Authors responded to all my comments and made necessary corrections. In my opinion, the current state of the manuscript allows its publication.